# One N-glycan regulates natural killer cell antibody-dependent cell-mediated cytotoxicity and modulates Fc γ receptor IIIa/CD16a structure

Paul G Kremer[1], Elizabeth A Lampros[1], Allison M Blocker[1], Adam W Barb[1,2,3]*

[1]Department of Biochemistry and Molecular Biology, University of Georgia, Athens, United States; [2]Complex Carbohydrate Research Center, University of Georgia, Athens, United States; [3]Department of Chemistry, University of Georgia, Athens, United States

## eLife Assessment

This study explores the mechanistic link between glycosylation at the N162 site of the Fc gamma receptor FcγRIIIa and the modulation of NK cell-mediated antibody-dependent cytotoxicity. Using innovative isotope labeling strategies and advanced NMR spectroscopy techniques, the authors provide **compelling** evidence of how glycan composition influences receptor stability and immune function. These findings offer **fundamental** insights that may contribute to the development of more effective therapeutic antibodies. The article will be of significant interest to immunologists and researchers focused on therapeutic antibody design.

*For correspondence:
adambarb@uga.edu

**Abstract** Both endogenous antibodies and a subset of antibody therapeutics engage Fc gamma receptor (FcγR)IIIa/CD16a to stimulate a protective immune response. Increasing the FcγRIIIa/IgG1 interaction improves the immune response and thus represents a strategy to improve therapeutic efficacy. FcγRIIIa is a heavily glycosylated receptor and glycan composition affects antibody-binding affinity. Though our laboratory previously demonstrated that natural killer (NK) cell N-glycan composition affected the potency of one key protective mechanism, antibody-dependent cell-mediated cytotoxicity (ADCC), it was unclear if this effect was due to FcγRIIIa glycosylation. Furthermore, the structural mechanism linking glycan composition to affinity and cellular activation remained undescribed. To define the role of individual amino acid and N-glycan residues, we measured affinity using multiple FcγRIIIa glycoforms. We observed stepwise affinity increases with each glycan truncation step, with the most severely truncated glycoform displaying the highest affinity. Removing the N162 glycan demonstrated its predominant role in regulating antibody-binding affinity, in contrast to four other FcγRIIIa N-glycans. We next evaluated the impact of the N162 glycan on NK cell ADCC. NK cells expressing the FcγRIIIa V158 allotype exhibited increased ADCC following kifunensine treatment to limit N-glycan processing. Notably, an increase was not observed with cells expressing the FcγRIIIa V158 S164A variant that lacks N162 glycosylation, indicating that the N162 glycan is required for increased NK cell ADCC. To gain structural insight into the mechanisms of N162 regulation, we applied a novel protein isotope labeling approach in combination with solution NMR spectroscopy. FG loop residues proximal to the N162 glycosylation site showed large chemical shift perturbations following glycan truncation. These data support a model for the regulation of FcγRIIIa affinity and NK cell ADCC whereby composition of the N162 glycan stabilizes the FG loop and thus the antibody-binding site.

## Introduction

Natural killer (NK) cells rapidly respond to destroy tissue coated with antibodies. This protective mechanism, termed antibody-dependent cell-mediated cytotoxicity (ADCC), is likewise exploited by many therapeutic monoclonal antibodies (mAbs) recognizing specific epitopes that escape detection by endogenous antibodies. However, NK cell ADCC elicited both by endogenous antibodies and mAbs fails to ameliorate disease in many patients. There is sufficient evidence to expect that increased NK cell ADCC will improve patient responses though few strategies exist to achieve this outcome.

NK cells require binding of antibodies to only a single receptor type, Fc gamma receptor IIIa (FcγRIIIa/CD16a), to elicit ADCC. Multiple lines of evidence support the role of increased antibody-binding affinity in improved NK cell responses. NK cells expressing the higher affinity FcγRIIIa V158 allotype demonstrate greater ADCC than those expressing the weaker-binding F158 allotype (hereafter referred to as V158F) (*Bruhns et al., 2009*; *Hayes et al., 2017*; *Cartron et al., 2002*; *Weng and Levy, 2003*). Furthermore, improving the FcγRIIIa-binding affinity of antibodies, which bind through the Fc region, likewise increases ADCC and therapeutic potency (*Presta et al., 2002*; *Mössner et al., 2010*; *Townsend et al., 2023*). Antibody engineering efforts have been widely explored to improve ADCC by increasing FcγRIIIa engagement; however, substantially less is known about the structural mechanisms that affect FcγRIIIa function.

Our laboratory identified an unexpected strategy to increase FcγRIIIa affinity. We determined that FcγRIIIa glycosylation affected antibody-binding affinity, with oligomannose-type N-glycans providing higher affinity interactions (*Patel et al., 2018*; *Subedi and Barb, 2018*). Oligomannose N-glycans are minimally remodeled forms that are not expected at high percentages on secreted proteins in contrast to highly remodeled complex-type glycans found on most serum glycoproteins (*Moremen et al., 2012*). Among the five FcγRIIIa N-glycosylation sites, the composition of the glycan at N162 is responsible for increased antibody-binding affinity and is located near the interface formed with the antibody Fc (*Subedi and Barb, 2018*; *Sondermann et al., 2000*). Furthermore, we identified a high degree of glycan compositional heterogeneity at the N162 site on FcγRIIIa purified from NK cells of healthy human donors including both complex-type and oligomannose glycoforms (*Patel et al., 2019*; *Roberts et al., 2020*; *Patel et al., 2020*). YTS cells, a key cytotoxic human NK cell line used for these studies, express FcγRIIIa with extensive glycan processing, including the N162 site with predominantly hybrid and complex-type glycoforms (*Patel et al., 2021*). The N162 glycan heterogeneity on NK cells is reflected in the presence of both high-affinity and low-affinity FcγRIIIa forms in the periphery of healthy donors, with high-affinity forms more abundant in adult donors compared to children (*Benavente et al., 2024*).

We likewise demonstrated that NK cell N-glycan processing reduced ADCC potency. NK cells with limited N-glycan remodeling capability, following either treatment with kifunensine or knockdown of the bottleneck glycan processing enzyme MGAT1, demonstrated increased ADCC (*Benavente et al., 2024*; *Rodriguez Benavente et al., 2023*). These studies, however, did not determine that FcγRIIIa N-glycan processing, nor the composition of the N162 glycan, mediated the increased ADCC. It is equally possible that this role is mediated by other NK cell glycans, of which hundreds are expected. Demonstration of the role of the N162 glycan in ADCC is expected to promote improved ADCC responses through efforts to tune glycan composition of endogenous NK cells or infused NK cells. Furthermore, such a demonstration is expected to promote FcγRIIIa engineering to increase ADCC beyond levels available from endogenous NK cells and naturally occurring FcγRIIIa variants.

Here we investigate the N162 glycan's role in FcγRIIIa structure, antibody-binding affinity and NK cell ADCC. We performed a mutational screen to identify FcγRIIIa residues that mediate antibody-binding affinity and sensitivity to the N-glycan composition. Introducing a subset of these FcγRIIIa variants, displaying a range of affinities, into cytotoxic NK cells provided insight into the relationship between antibody-binding affinity and ADCC. We then defined the impact of N162 glycan composition on NK cell ADCC and FcγRIIIa structure. As a result, we define a link between the N162 glycan composition and NK cell ADCC.

## Results

### FcγRIIIa affinity impacts ADCC

We generated a library of FcγRIIIa proteins with mutations at the antibody-binding interface to identify key residues. The resulting FcγRIIIa proteins revealed a wide range of affinities between those slightly higher than wildtype (S164A) to a handful with no apparent binding at 2.5 µM to fucosylated IgG1 Fc (Y132S, W90A, W113A; *Figure 1A*, *Supplementary file 1*). Mapping the impact of these mutations onto a FcγRIIIa structural model identified two key regions consistent with empirically derived structural models of the antibody:receptor complex (*Sondermann et al., 2000*), though our data provide greater detail showing how each residue affects affinity. These two regions center around Y132 and a pair of tryptophan residues (W90 and W113), which overlaps with the contact interface identified by X-ray crystallography (*Figure 1B*). Surprisingly, although the K161A mutation reduces affinity by tenfold, mutating adjacent residues including S160 showed little direct impact on affinity. It was likewise surprising that the solvent-exposed, large hydrophobic side chains of F153 and I88 minimally impacted affinity (less than twofold). A majority of the mutations only impacted affinity by threefold or less.

The FcγRIIIa V158 allotype (hereafter referred to as the wildtype) binds approximately fourfold tighter than the more common V158F allotype and NK cells expressing the wildtype likewise exhibit greater ADCC (*Bruhns et al., 2009*; *Vidarsson et al., 2014*); however, beyond these two points the relationship between FcγRIIIa affinity and ADCC remains undefined (*Kremer and Barb, 2022*). A positive correlation between affinity and ADCC would indicate improved NK cell-mediated therapies may result from increasing the FcγRIIIa antibody-binding affinity beyond that of the wildtype. This FcγRIIIa library evaluated above, plus a range of other mutations proximal to this interface, provides a wide range of FcγRIIIa affinities (the complete binding affinities of all FcγRIIIa variants are shown in *Supplementary file 1*).

We next evaluated the impact of FcγRIIIa mutations on ADCC potency. We transduced cytotoxic YTS NK cells to express a subset of FcγRIIIa variants from our affinity screen, encompassing a broad affinity range, using lentivirus to establish stable cytotoxic NK cell lines. Each YTS cell line displayed comparable FcγRIIIa expression, however, at levels somewhat reduced from the original YTS-FcγRIIIa cell line provided by Dr. Mace (Columbia U., *Figure 1—figure supplement 1*). A comparison of the original YTS-FcγRIIIa (V158) cells and our transduced YTS cell line expressing the same wildtype receptor showed greater expression and greater ADCC from the original YTS-FcγRIIIa cell line likely resulting from our cell isolation strategy that selected only for GFP+ cells rather than FcγRIIIa expression. Within our library, YTS cells expressing the FcγRIIIa S164A variant displayed the highest average ADCC and the tryptophan mutants showed no detectable ADCC, indicating higher affinity leads to higher ADCC (*Figure 1C*). A plot of the affinity for each variant versus the ADCC potency demonstrated a trend linking high affinity and greater ADCC. These results indicate that greater FcγRIIIa antibody-binding affinity on NK cells increases ADCC and FcγRIIIa engineering represents an unexplored avenue to improve NK cell-mediated immunotherapies.

### The FcγRIIIa N162-glycan is required for high-affinity interactions with afucosylated IgG

Because affinity is a key determinant of ADCC potency, we further explored FcγRIIIa changes that affect affinity and to identify variants with altered sensitivity to IgG1 Fc glycan composition. Shields et al. previously showed that antibody fucosylation reduces affinity for FcγRIIIa (*Shields et al., 2002*), and it is known that the N162 glycan mediates the response to fucose, though the underlying mechanism remains disputed (*Ferrara et al., 2011*; *Mizushima et al., 2011*; *Sakae et al., 2017*; *Falconer et al., 2018*). We identified FcγRIIIa residues within the contact distance of the N162 glycan to explore the impact of proximal residues to antibody fucosylation. We next evaluated mutations at each contact site using both the wildtype and S164A variant that prevents N162 glycosylation. These variants largely exhibited decreased affinity compared with the wildtype FcγRIIIa when evaluated against fucosylated IgG1 Fc (*Figure 2A*). Surprisingly, each variant displayed reduced sensitivity to fucose when the N162 glycan was absent. The impact of fucose reduced from a 2.2-fold average effect when the N162 glycan was present to a 1.3-fold change when the N162 glycan was absent (*Figure 2B*). Most variants bound afucosylated IgG1 Fc with weaker affinity. The F153S and R155S variants lacking the N162 glycan are notable outliers in that both bind antibody glycoforms with higher affinity than the wildtype.

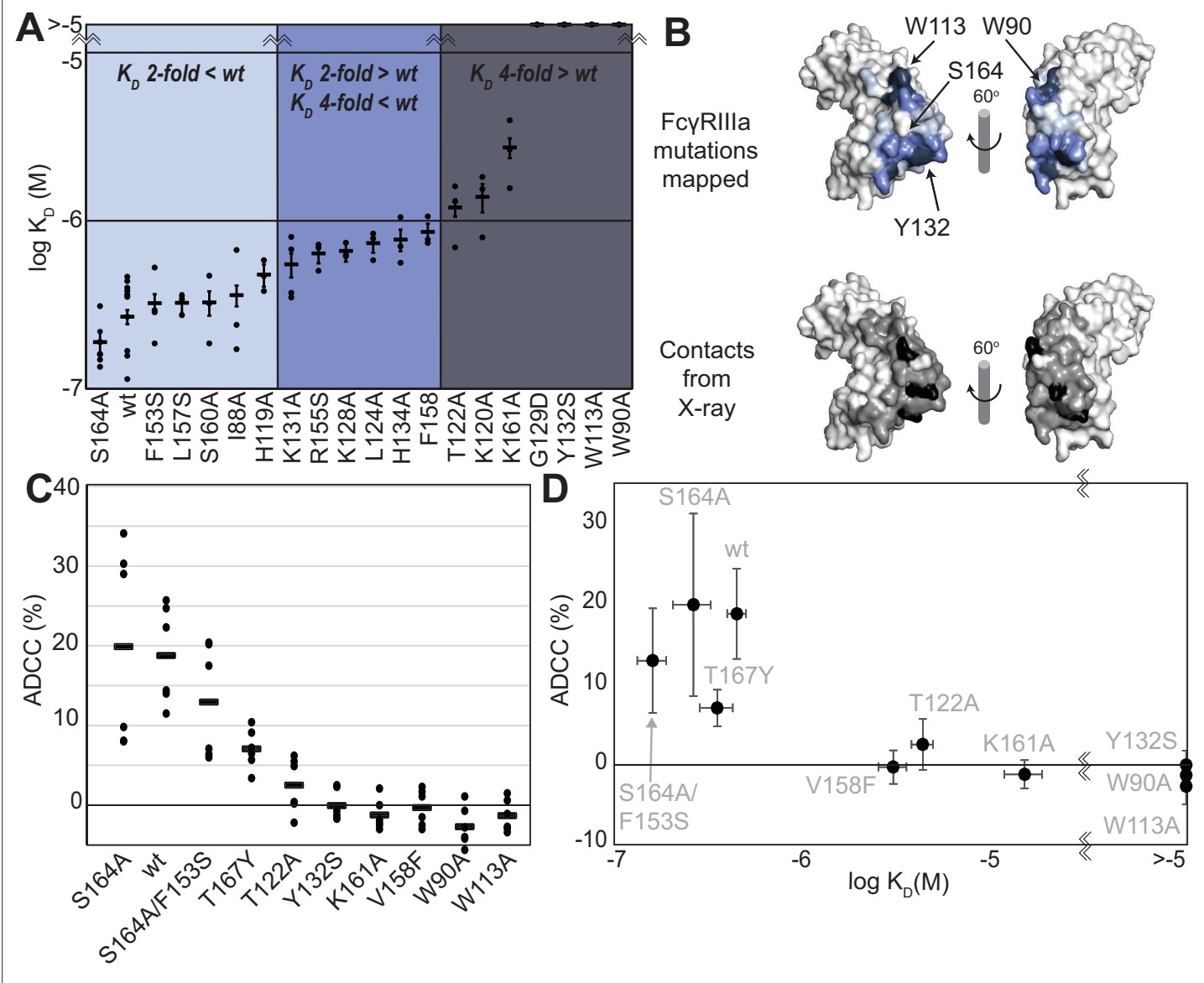

**Figure 1.** FcγRIIIa antibody-binding affinities correlate with antibody-dependent cell-mediated cytotoxicity (ADCC) potency. (**A**) Affinities of FcγRIIIa amino acid variants determined by SPR are binned into impact: twofold lower (light blue), between two- and fourfold (blue), or greater than fourfold lower than V158 (dark blue). (**B**) Plotting these values onto a surface representation of FcγRIIIa, using the same coloring scheme as in (**A**)., reveals two critical areas for binding centered around W113 and Y132 (top panel). These values provide more detail in contrast to the interface defined by X-ray crystallography with contacts shown <3 Å (black surface) and <5 Å (gray surface). (**C**) ADCC of YTS cells transduced to express a panel of FcγRIIIa variants. Horizontal black bars represent average ADCC values, with individual point representing individual assays. Experiments were completed in triplicate and the figure includes data from multiple experiments collected on multiple days. (**D**) A comparison of ADCC values from panel (**C**) and binding affinity from panel (**A**) shows a correlation. Scale bars in D represent n=6 ADCC assays and n>=3 affinity measurements.

The online version of this article includes the following source data and figure supplement(s) for figure 1:

**Figure supplement 1.** Comparison of expression and antibody-dependent cell-mediated cytotoxicity (ADCC) of the previously established YTS-FcγRIIIa natural killer (NK) cell line and the lentivirus-transduced cell lines prepared herein.

**Figure supplement 1—source data 1.** Original image file containing the original west blots for *Figure 1—figure supplement 1*.

**Figure supplement 1—source data 2.** Image file containing the original west blots for *Figure 1—figure supplement 1*, indicating the relevant bands and treatments.

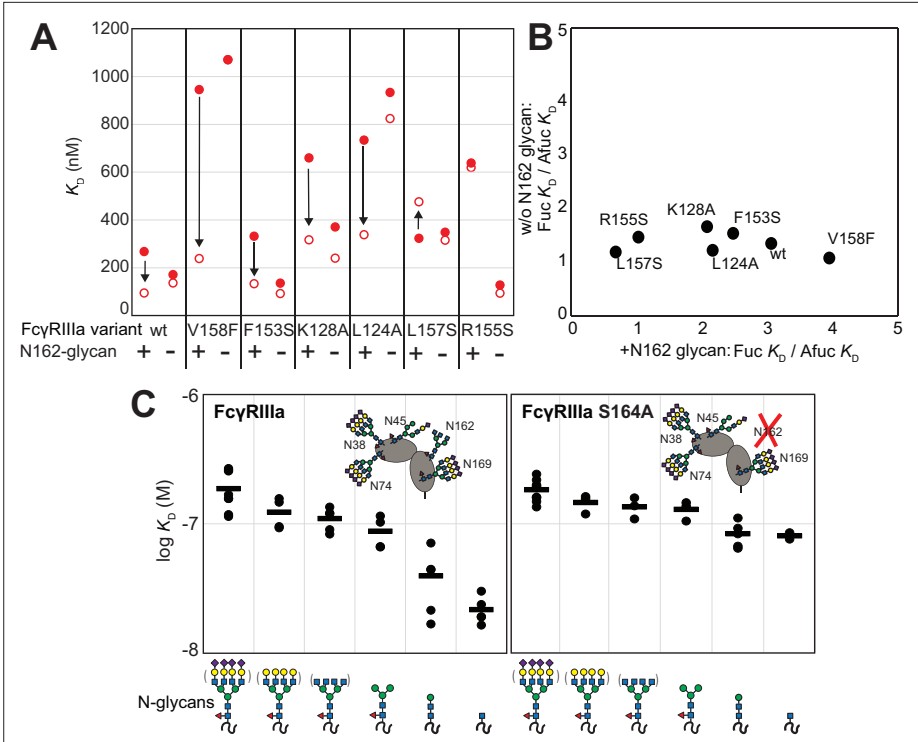

**Figure 2.** The FcγRIIIa N162 glycan regulates affinity toward fucosylated and afucosylated IgG1 Fc. (**A**) FcγRIIIa variants demonstrate higher affinity toward without IgG1 Fc core-fucosylation (open circles) than with this modification (red circles). The affinity increase is demarcated with a vertical arrow. When the N162 glycan was removed through the S164A mutation, the fucose sensitivity greatly diminished. (**B**) Comparison of the fold affinity increase in panel (**A**) due to removing IgG1 Fc fucose. Averages for the fold increase with the N162 glycan present and absent are noted on the x- and y-axes, respectively. (**C**) The affinities of six different FcγRIIIa glycoforms were measured with and without the N162 glycan (wt and S164A, respectively). Horizontal black bars represent the means and individual measurements are shown with closed black circles. Cartoon models utilize the SNFG nomenclature and represent the possible N-glycan compositions for each species.

The online version of this article includes the following source data and figure supplement(s) for figure 2:

**Figure supplement 1.** Glycosidase digestions visualized by SDS-PAGE.

**Figure supplement 1—source data 1.** Original image file containing the SDS-PAGE analysis for *Figure 2—figure supplement 1*.

**Figure supplement 1—source data 2.** Image file containing the SDS-PAGE analysis for *Figure 2—figure supplement 1*, indicating the relevant bands and treatments.

These data also are consistent with the hypothesis that intermolecular glycan-glycan contacts provide minimal stability to the antibody/receptor complex because higher affinity was achieved by removing the N162 glycan.

## N-glycan truncations reveal how each residue contributes to affinity

In addition to glycan composition of the IgG1 Fc ligand affecting affinity, composition of the FcγRIIIa glycans likewise impacts antibody-binding affinity (*Patel et al., 2018*; *Subedi and Barb, 2018*). It is tempting to conclude from the preceding study that higher affinity interactions can be achieved by removing the N162 glycan entirely; however, our previous work showed that truncating the glycan to a single (1)GlcNAc residue provides superior affinity though it was unclear if these changes mapped to the N162 site (*Kremer and Barb, 2022*). It is also possible that glycoforms slightly longer than the (1)GlcNAc truncated form provide even greater affinity though this possibility remained untested. To identify which N-glycan residues affected affinity, we treated recombinant FcγRIIIa with a series of glycosidases to sequentially remove terminal sugars from the nonreducing end (*Figure 2—figure supplement 1*). We observed a step-wise increase in affinity following the removal of each type of

terminal sugar (*Figure 2C*, left panel). Removing the core α-linked mannose or sialic acid residues showed the greatest change for a single removal step, increasing affinity by fivefold or twofold respectively. Overall, glycan truncation was observed to decrease $K_D$ by 12-fold. To evaluate the individual impact of the N162 glycan, we next truncated glycans of the S164A variant that lacks N162 glycosylation but retains four other N-glycans (*Figure 2B*, right panel). While the S164A affinities followed the same trend as the wildtype, the changes were substantially smaller in magnitude with only a fourfold overall change. The removal of the core α-linked mannose residues increased affinity by less than 1.5-fold, in contrast to the fivefold increase observed for the wildtype. These results demonstrate the primary contribution of the N162 glycan composition to antibody-binding affinity, with smaller contributions from N-glycans at the remaining four sites. Furthermore, these data demonstrate the negative impact of adding additional residues throughout the processing pathway, further indicating that N-glycan processing inhibits antibody-binding affinity.

## The FcγRIIIa N162 glycan is required for increased ADCC potency following kifunensine treatment

The data above demonstrate the impact of FcγRIIIa N162 glycan composition on antibody-binding affinity. Furthermore, our lab previously showed that inhibiting NK cell glycan processing, and thus FcγRIIIa processing, increased ADCC potency (*Benavente et al., 2024*; *Rodriguez Benavente et al., 2023*). However, these studies did not determine that composition of the FcγRIIIa N162 glycan affected ADCC potency. We expect that the FcγRIIIa N162 glycan composition that was shown to mediate high-affinity interactions with IgG1 Fc likewise is required for the potent ADCC observed following blockage of NK cell N-glycan processing (*Subedi and Barb, 2018*; *Rodriguez Benavente*

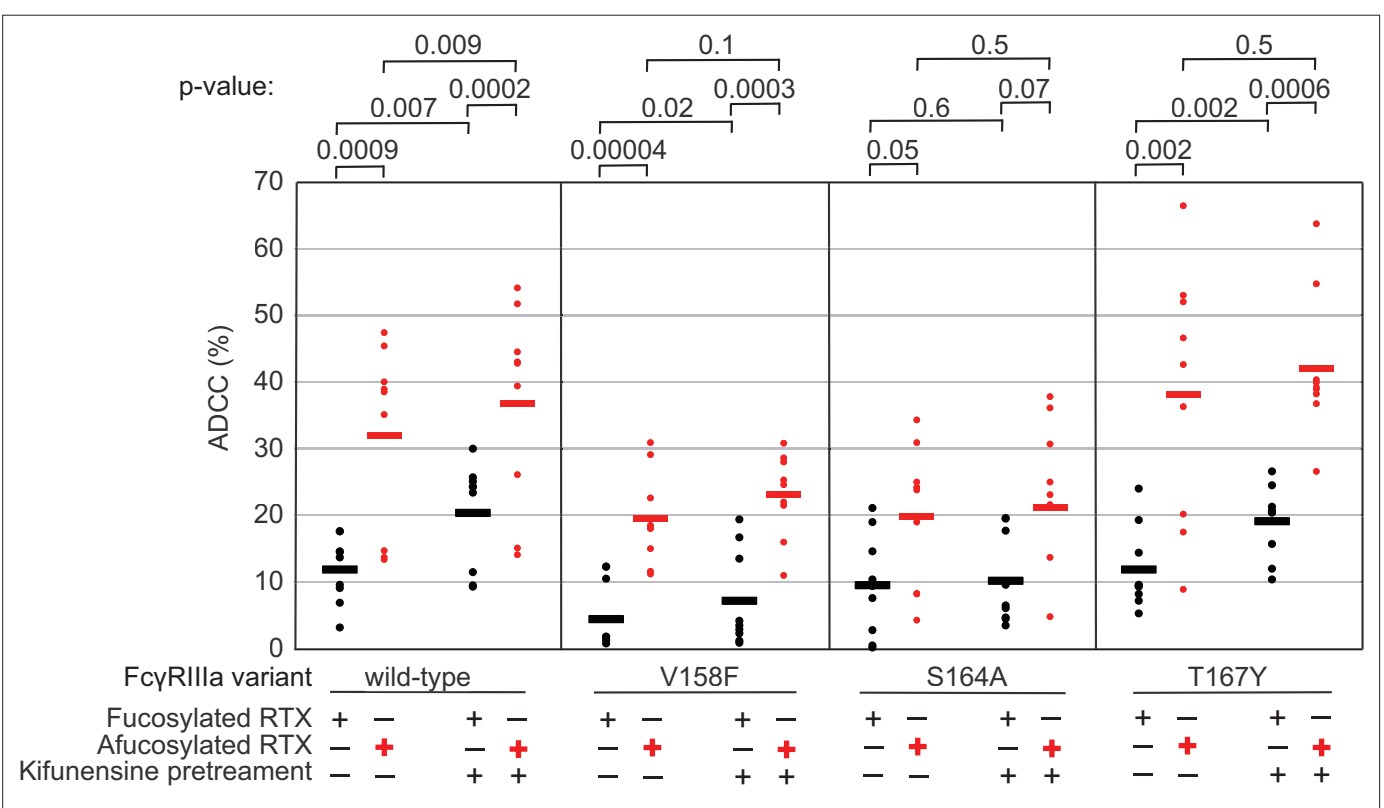

**Figure 3.** The FcγRIIIa N162 glycan regulates natural killer (NK) cell antibody-dependent cell-mediated cytotoxicity (ADCC). The ADCC of NK cells increases significantly following 20 μM kifunensine for YTS cells expressing FcγRIIIa that retains the N162 glycan (wildtype, V158F, T167Y). Removing the N162 glycan with the S164A mutation eliminates this increase. In addition to kifunensine, these cells demonstrate significant ADCC increases from afucosylated rituximab (RTX) compared to fucosylated RTX. The YTS cells FcγRIIIa S164A cells likewise demonstrate no increased ADCC following kifunensine treatment when using afucosylated RTX, unlike YTS cells expressing the wildtype FcγRIIIa. Observations made using an afucosylated antibody are shown in red. Data shown include three independent experiments collected on three different days, each with three replicates (n=9). p-Values from two-tailed *t*-tests are shown at the top. Raw ADCC values supporting this figure are presented in *Supplementary file 2*.

*et al., 2023*). To evaluate the role of N162 glycosylation in NK cell ADCC, we treated our FcγRIIIa-expressing YTS NK cell lines from *Figure 1C* with kifunesine, an inhibitor of N-glycan processing that specifically reduced YTS cell N-glycan processing (*Benavente et al., 2024*). As expected based on our lab's previous work, the YTS cells expressing the FcγRIIIa wildtype demonstrated increased ADCC following kifunesine treatment (*Figure 3*). Cells expressing the V158F variant likewise demonstrated a 1.4-fold ADCC increase following kifunensine treatment. We observed a similar result for YTS cells expressing the T167Y variant. However, YTS FcγRIIIa cells lacking only the N162 glycan through the S164A mutation demonstrated no increase following kifunensine treatment. Notably, the wildtype, V158F and T167Y variants all contain the N162 glycan but only S164A does not; thus, the N162 glycan is required for increased ADCC potency following kifunensine treatment.

We next evaluated afucosylated antibodies in combination with kifunensine in our cell lines, having previously demonstrated afucosylated antibodies acted synergistically with kifunensine treatment to increase NK cell ADCC (*Rodriguez Benavente et al., 2023*). The YTS cells expressing wildtype FcγRIIIa evaluated with afucosylated antibody showed a significant increase following kifunensine treatment compared to those evaluated with the fucosylated antibody (*Figure 3*). Notably, we observed no further benefit for the S164A variant with afucosylated antibodies. This result indicates that the N162 glycan is a critical mediator of increased ADCC responses following inhibiting both NK cell glycan remodeling and antibody fucosylation.

## FcγRIIIa backbone resonance assignment

Having demonstrated the role of the FcγRIIIa N162 glycan in antibody-binding affinity and NK cell ADCC, we next evaluated how glycan composition affects FcγRIIIa structure to identify regions of the protein that modulate affinity. We probed the effect of N-glycan composition using solution NMR spectroscopy, a technique that evaluates FcγRIIIa structure in dilute solutions and can identify the impact of highly mobile elements including disordered loops, N and C termini, and glycans. First, individual peaks in the NMR spectra must be assigned to unique atoms in the protein to evaluate structural changes with atomic resolution. N-glycoproteins, including FcγRIIIa, represent a significant challenge for NMR spectroscopy due to exceptional line broadening and thus signal loss resulting from the presence of extended N-glycans that slow molecular tumbling. As a result of these physical limitations, N-glycoproteins are seldom investigated by NMR despite their prevalence in the human proteome (22% of the human proteome enters the secretory pathway and contains an N-glycosylation site). Broad spectral lines dramatically limit the quality of multidimensional spectra required for the standard resonance assignment strategies, preventing the collection of CB frequencies that provide a high degree of amino acid discrimination.

To surmount these limitations, we first increased tumbling and thus spectral quality by mutating three of the five N-glycosylation sites, retaining N-glycans at N45 and N162 that stabilize antibody binding and solubility (*Subedi and Barb, 2018*). The spectrum of the FcγRIIIa N38Q/N74Q/N169Q variant with two N-glycans improved and peak positions proved highly comparable to those from FcγRIIIa with five N-glycans. Next, we simplified the assignment problem by defining the amino acid type for many peaks in the spectrum through specific amino acid labeling. Our strategy supplements the HEK293F growth medium with individual [$^{15}$N]-labeled amino acids (*Subedi et al., 2024*) and provided residue information for 13 types (C,F,G,H,I,K,N,R,S,T,V,W,Y; an example spectrum is shown in *Figure 4—figure supplement 1*). These residue-type assignments guided the definition of residue connections, and thus the backbone assignment, in three-dimensional HNCA and HN(CO)CA spectra. Carbonyl carbon resonances were then assigned using an HNCO experiment. As a result, we assigned a high percentage of the backbone resonances (87% N, 90% HN, 86% CO, 86% CA) and almost all the peaks in the 2D HSQC-TROSY spectrum (*Figure 4A and B*). This resonance assignment is suitable for interpreting structural changes associated with changing the N-glycan composition and represents a significant triumph due to the challenges associated with glycoprotein NMR. To our knowledge, this is the only NMR resonance assignment of a glycoprotein with two N-glycans attached. A small handful of assignments exist for proteins with one N-glycan, notably CD2 and the Fc region of IgG1 (a thermostable dimer) (*Wyss et al., 1995*; *Yagi et al., 2015*).

A comparison of the peak positions between the FcγRIIIa assignment and a previous assignment of the similar FcγRIIIb protein reveals a high degree of overlap but a few notable differences (*Figure 4C*). These two proteins differ by eight amino acids, though FcγRIIIb was expressed in *Escherichia coli* and

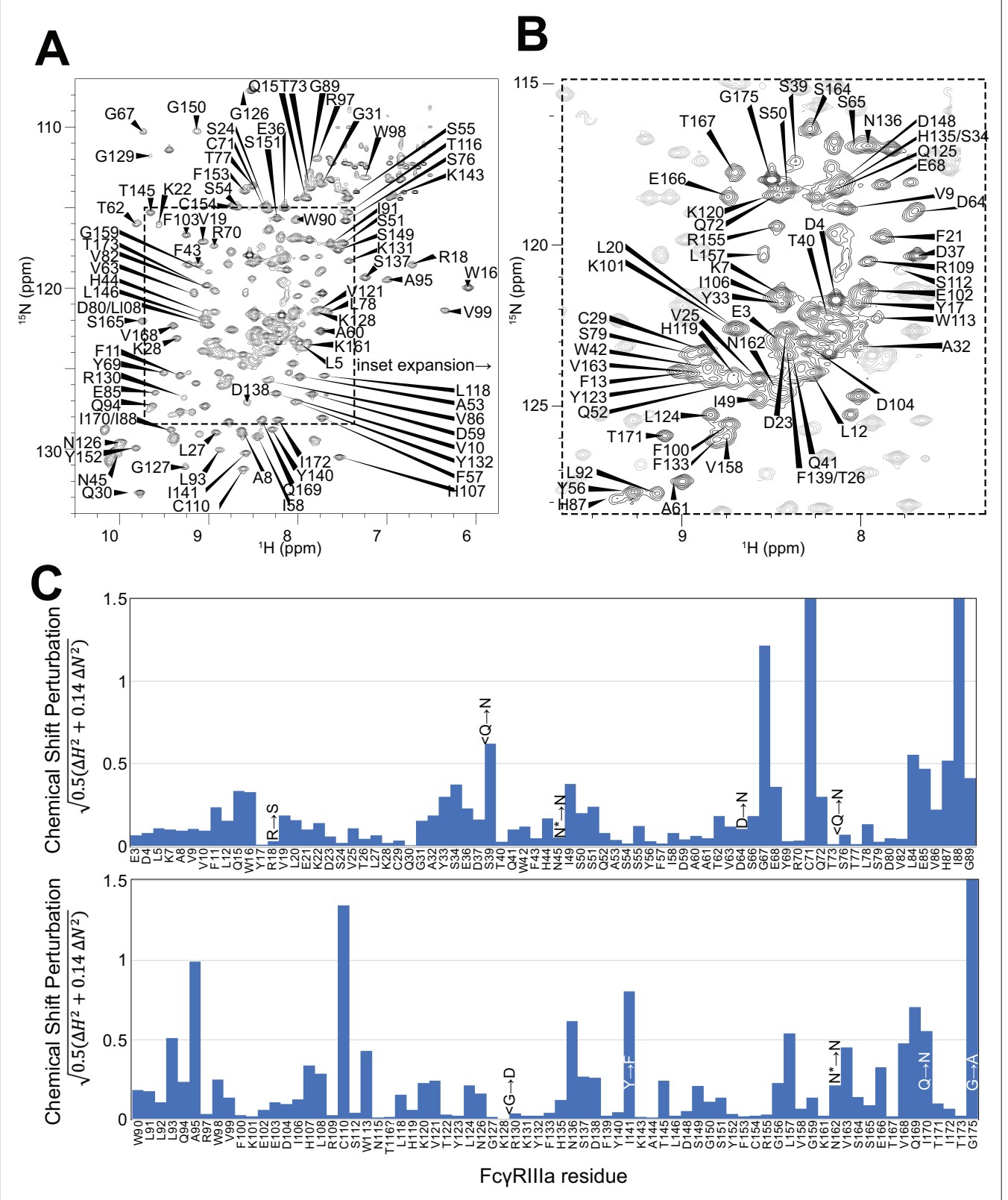

**Figure 4.** Backbone resonance assignment of FcγRIIIa with N-glycans at N45 and N162. Amino acid position within the sequence of 175 residues and the residue type are shown. (**A**) The entire ¹H-¹⁵N HSQC-TROSY spectrum. (**B**) Expansion and additional assignments within the inset. (**C**) A comparison of the assigned ¹H and ¹⁵N resonances from the glycosylated FcγRIIIa to FcγRIIIb expressed from *E. coli* that contains no N-glycans. Sequence differences are noted in the figure, with the bottom letter denoting the FcγRIIIa residue. N* = a glycosylated asparagine residue.

*Figure 4 continued on next page*

*Figure 4 continued*

The online version of this article includes the following figure supplement(s) for figure 4:

**Figure supplement 1.** Residue-specific labeling of the glycosylated FcγRIIIa.

**Figure supplement 2.** Strip plots from triple resonance HNCA and HN(CO)CA experiments showing the assignments of the C71 and C110 resonances.

thus is not glycosylated (*Yogo et al., 2018*). Two notable differences occur at C71 and C110 that are distinguished in the HNCA and HN(CO)CA spectra support this assignment (*Figure 4—figure supplement 2*). Surprisingly, the N-glycosylation of the N45 and N162 residues revealed minimal impact on the backbone peak positions for these asparagine residues. A few other differences emerged in regions near amino acid differences, notably position 18 and the contact region of ~80–99, as well as 38, 129 140, and 169. We previously defined local structural differences resulting from substitution at position 129 using X-ray crystallography that are consistent with this NMR-based analysis (*Roberts and Barb, 2018*).

## FcγRIIIa N-glycan composition affects the structure of the antibody-binding site

We applied our protein labeling approach and backbone resonance assignment to define individual FcγRIIIa atoms that experience different environments based on N-glycan composition. We expressed three FcγRIIIa glycoforms with six [$^{15}$N]-labeled amino acids to introduce probes throughout the polypeptide (*Figure 5A*, *Figure 5—figure supplement 1*). The spectra showed dispersed peaks that revealed differences between different glycoforms (*Figure 5C*). A few residues showed evidence for the sampling of multiple conformations, notably V158 and K161. These residues are located on the FG loop that contains the N162 glycan. The peaks simplify as the glycan becomes shorter, up to the point of a single peak observed for V158 with the (1)GlcNAc glycoform that may indicate either sampling of a single conformation or more rapid exchange between two or more conformations.

Mapping the chemical shift perturbations to the primary sequence revealed residues proximal to the five N-glycosylation sites when comparing the receptor with complex-type glycans to that with Man5 N-glycans (*Figure 5D and E*). Truncation to the (1)GlcNAc glycoform caused greater changes to the regions around the N45 and N162 glycans (*Figure 5D and F*). These data likely indicate a change in the FcγRIIIa conformation sampled by residues near the FG loop, and potentially changes in the orientation between the two extracellular domains as chemical shift differences likewise map to regions near the connection between these two domains. Features associated with the (1)GlcNAc spectra represent conformations that promote higher affinity interactions as this glycoform binds antibody with the greatest affinity of those evaluated. We anticipate that these conformations are sampled by these residues on the surface of NK cells, and the conformational sampling on the cell surface is dictated by the N162 glycan composition.

## Evidence supporting distinct conformations of the FcγRIIIa FG loop

Solution NMR evidence identified differential FcγRIIIa conformational sampling, particularly residues in the FG loop, and is supported by the appearance of distinct conformations identified by X-ray crystallography. A recent structure isolated one conformation of the FG loop that differed from those previously observed, mostly through structures bound to IgG1 Fc (*Sondermann et al., 2000*; *Falconer et al., 2018*; *Kakiuchi-Kiyota et al., 2022*). In this recent structure of glycosylated FcγRIIIa, the FG loop is free from contacts with the Fc ligand or with the crystal lattice, revealing a conformation that is sterically prohibited in the complex with antibody (*Figure 6A*). To bind, the N162 glycan and the FG loop must reorient to avoid steric clashes with IgG1 Fc. The backbone conformation of the supporting beta strands shows minimal displacement, though larger backbone deviations are observed for the L157, K161 and S164 residues (*Figure 6—figure supplement 1*). The presence of these distinct conformations is supported by the NMR spectra showing two peaks, including V158 and K161 (*Figure 5C*). We investigated conformational sampling further with 1 µs all-atom MD simulations and found the presence of both conformations identified by X-ray crystallography in trajectories of FcγRIIIa with complex-type or Man5 N-glycans (*Figure 6B*). These simulations were initialized with a starting structure representing the bound state conformation but sampled both conformations throughout the experiment. One notable detail from these simulations was the formation of

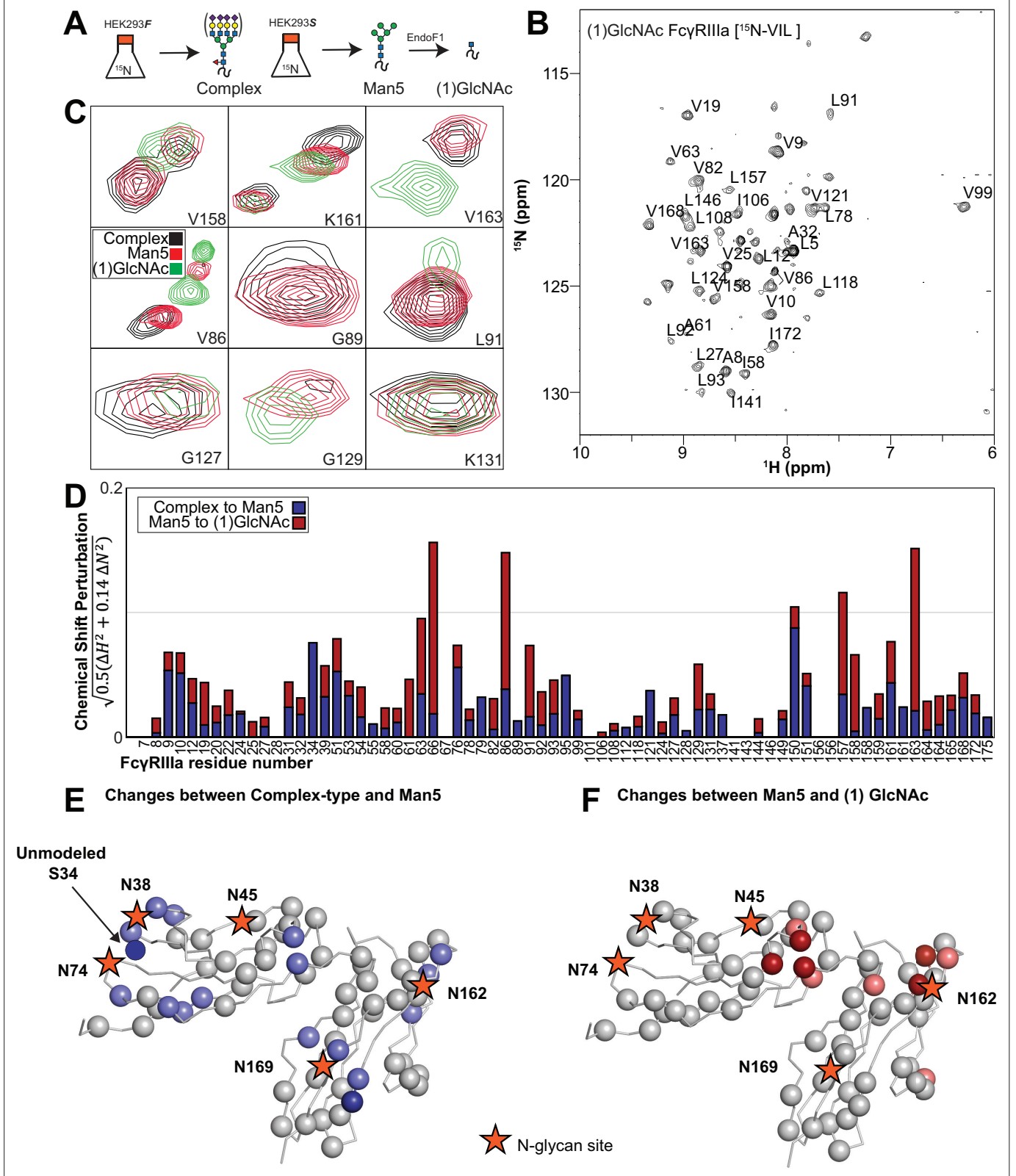

**Figure 5.** Glycan composition changes the FcγRIIIa backbone conformation. (**A**) Diagram of the protein expression, labeling, and glycan remodeling procedures. (**B**) Example HSQC-TROSY spectrum of FcγRIIIa with the truncated (1)GlcNAc N-glycan labeled with ¹⁵N-(Val,Leu,Ile) during expression. (**C**) Isolated peaks show differences in position between different glycoforms. (**D**) The observed Chemical Shift Perturbation (CSP) between complex-type

*Figure 5 continued on next page*

*Figure 5 continued*

and Man5 N-glycans (blue) or Man5 and (1)GlcNAc N-glycans (red) is shown by residue number. (**D, E**) CSPs > 0.03 (light) and >0.06 (dark) mapped to a ribbon model of FcγRIIIa. Truncation to (1)GlcNAc causes CSPs near the Fc-binding interface that is proximal to N162.

The online version of this article includes the following figure supplement(s) for figure 5:

**Figure supplement 1.** Overlayed HSQC-TROSY spectra of [15]N-labeled FcγRIIIa with complex-type (black) 5-mannose (red) or (1)GlcNAc (green) N-glycans.

non-covalent interactions between the aliphatic portion of the R155 sidechain with the hydrophobic surface of the (1)GlcNAc residue in the ligand-bound conformation that were absent in the unliganded conformation. To evaluate the importance of this interaction, we measured the binding affinity of the truncated FcγRIIIa (1)GlcNAc glycoform lacking the R155 sidechain. The FcγRIIIa R155S (1)GlcNAc glycoform bound IgG1 Fc with a fivefold weaker affinity compared to the wildtype in the same glyco-form (*Figure 6C*, *Supplementary file 1*). The R155S binding is comparable to the S164A mutant

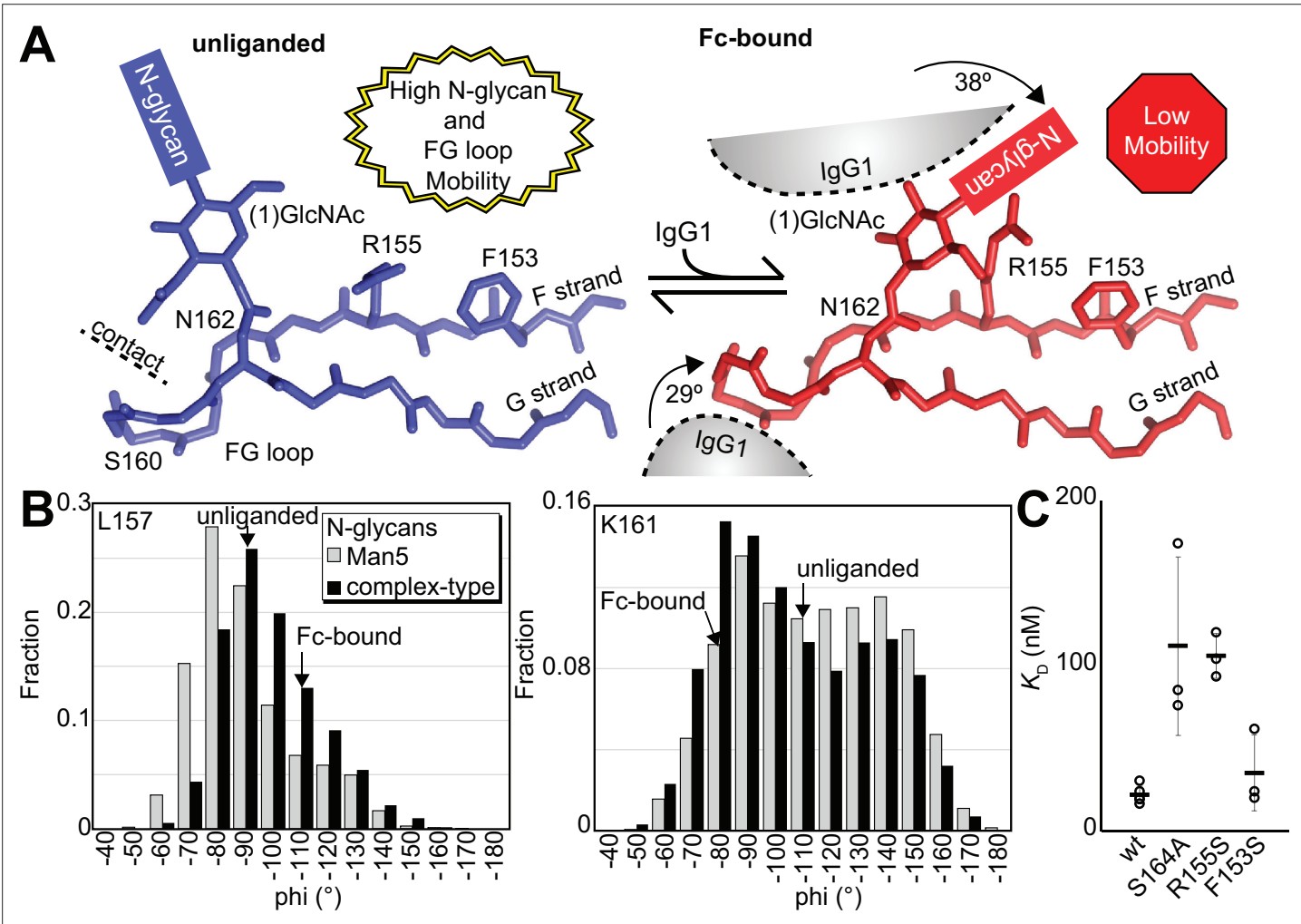

**Figure 6.** Binding antibody induces an FcγRIIIa conformational change. (**A**) Two conformations of the FG loop previously captured by X-ray crystallography (pbd 7seg, 5vu0). Both the FG loop and the N-glycan become restricted to accommodate IgG1 Fc. The conformational entropy of a complex-type N-glycan, with more rotatable bonds, is greater as is the loss of entropy upon binding compared to smaller N-glycans. (**B**) Evidence for conformational sampling in the unliganded FcγRIIIa revealed by all-atom molecular dynamics simulations. Each data set shows the average of two independent 1 μs trajectories, with separate experiments for FcγRIIIa with Man5 N-glycans or complex-type N-glycans. (**C**) Binding affinity of FcγRIIIa variants following EndoF1 digestion, displaying a truncated (1)GlcNAc N-glycan. Scale bars represent errors for n=3 measurements.

The online version of this article includes the following figure supplement(s) for figure 6:

**Figure supplement 1.** Backbone dihedral angles from two FcγRIIIa X-ray crystallography structures.

lacking the N162-glycan, and mutation of a nearby residue retaining the N-glycan F153S retains the benefit of the (1)GlcNAc/R155 interaction. These data demonstrate a role for the R155 sidechain to stabilize the binding-competent conformation.

## Discussion

### The N162 glycan regulates NK cell ADCC

Removing the N162 glycan eliminated the ADCC gains following inhibiting NK cell N-glycan processing with kifunensine, demonstrating a unique role for this protein modification in the immune system. These data demonstrate why NK cell ADCC is sensitive to changes of the N-glycan composition, an effect that was previously shown to require antibody binding to FcγRIIIa (*Rodriguez Benavente et al., 2023*). This result is likewise consistent with prior binding affinity measurements defining a role for N162 glycan composition as a factor in antibody-binding affinity (*Patel et al., 2018*; *Subedi and Barb, 2018*). The binding affinity differences due to glycan composition are supported through studies of both human FcγRIIIa and macaque FcγRIII (*Hayes et al., 2017*; *Van Coillie et al., 2022*; *Tolbert et al., 2022*). The simple observation that the N162 glycan alone accounts for the sensitivity of differences in NK cell ADCC due to N-glycan processing is belied by the potentially enormous number of alternative hypotheses that ADCC is influenced by any other N-glycan on the NK cell surface. However, a distinction can be made between these explanations because of the simple difference of a single S164A mutation that eliminated the N162 glycosylation site. The specificity of this result defines a role for N162 glycan composition in NK cell ADCC.

We propose that the N162 glycan regulates NK cell ADCC, representing a previously undescribed regulatory element. The appearance of endogenous heterogeneity of the N162-glycan supports the assertion. The N162 glycan composition isolated from peripheral NK cells is notably heterogeneous, in contrast to the four remaining FcγRIIIa N-glycans that show remarkable similarity between different donors and cell types (*Patel et al., 2019*; *Roberts et al., 2020*; *Patel et al., 2020*). Thus, the FcγRIIIa N162 glycan composition is variable on endogenous NK cells, likely providing a range of ADCC potency. It remains unclear how cells produce variability restricted to the N162 N-glycan, with a high degree of similarity at the remaining four sites on the same protein despite the physical linkage that exposes these glycans to comparable conditions during secretion. Identifying this mechanism that affects N162 glycan processing has the potential to impact cellular immunotherapies if endogenous processing can be tuned in vivo. Furthermore, the advent of infused NK cell-based therapies provides a mechanism to deploy FcγRIIIa engineered with greater affinity or altered sensitivity to glycan composition of the expressing cell of the antibody ligand (*Page et al., 2024*). Removing the N162 glycan with the S164A mutation removes variability related to NK cell glycan processing, a potential confounding but unexplored factor in NK cell culture. Removing this glycan likewise reduces the impact of afucosylated antibodies on ADCC, which may be preferable if pathogenic afucosylated antibodies are present or enhanced ADCC is not desirable. Afucosylated antibodies have been reported at high titers during various autoimmune or viral diseases (*Larsen et al., 2021*; *Kapur et al., 2014*; *Ackerman et al., 2013*).

### Relationship between FcγRIIIa affinity and ADCC

Multiple lines of evidence indicate ADCC responses improve with antibodies engineered to engage FcγRIIIa with improved affinity; however, much less information is available on how improved FcγRIIIa antibody-binding affinity affects ADCC. This information is important to predict how future affinity improvements will impact ADCC. It is not obvious that increased antibody-binding affinity would promote greater ADCC considering the highly multivalent nature of the NK cell interaction with an opsonized target cell. Here we quantified ADCC using NK cells expressing a range of FcγRIIIa affinities outside the range defined by the natural V158 and V158F allotypes. Affinities lower than 300 nM showed little, if any, ADCC. Though donor NK cells displaying the V158F allotype exhibit measurable ADCC (*Hatjiharissi et al., 2007*), our reduced ADCC is likely due to the lowered FcγRIIIa expression, as noted in *Figure 1—figure supplement 1*. Affinities greater than 300 nM showed substantial cytotoxicity with a sharp increase at greater affinities indicating that greater FcγRIIIa antibody-binding affinity is expected to further increase ADCC. Notably, the ADCC potency for those high-affinity variants does not fall cleanly on a line, indicating that other factors affect our observations, which may

include organization at the cell surface, changes to glycan composition, or receptor trafficking. Thus, these data define a positive relationship between FcγRIIIa affinity and ADCC.

## A structural mechanism linking N162 composition to increased affinity

Analysis of unliganded and Fc-bound FcγRIIIa reveals structural differences proximal to N162 that are supported by the appearance of multiple conformations in NMR spectra and MD simulations. Both the FG loop and its attached N162 glycan move from a position in the unliganded state that allows maximum flexibility to a restricted state to accommodate Fc binding (*Figure 6A*). Furthermore, the sequential truncation of glycan residues provides increasingly greater affinity that is due largely to the N162 glycan (*Figure 2C*). We believe this result is best explained by a conformational entropy penalty introduced upon complex formation by shifting the N162 glycan to a restricted space between surfaces formed by the Fc and FcγRIIIa polypeptide. An alternative hypothesis, analogous to IgG1 Fc with intramolecular interactions stabilizing the N-glycan to increase affinity, appears to be less suitable for the FcγRIIIa data presented here largely because FcγRIIIa affinity increases following glycan truncation unlike IgG1 Fc N-glycan truncation that decrease affinity (*Subedi et al., 2014*; *Subedi and Barb, 2015*). Indeed, previous MD simulations likewise demonstrated a loss of N162 glycan conformational heterogeneity upon binding (*Falconer et al., 2018*).

A loss of N162 glycan conformational entropy alone is sufficient to explain the observed 1.7–2.2 kcal mol$^{-1}$ difference in binding affinities (*Subedi and Barb, 2018*). A simple calculation, assuming three equally populated conformations in the unliganded state per rotatable bond, provides an estimate of the difference between the change of conformational entropy upon binding for the complex-type and Man5 N-glycans of 4.8 kcal mol$^{-1}$ (*Imberty et al., 1993*). This value likely overestimates the penalty, being based on sampling only one N-glycan conformation in the Fc-bound state and notably excludes solvent entropy. Prior ITC analyses of these interactions revealed enthalpy/entropy compensation that precluded a clear definition of the contribution from conformational entropy from those data alone (*Falconer et al., 2018*). The entropic penalty is likely also alleviated by stabilizing interactions between the (1)GlcNAc residue and the R155 sidechain, explaining why the truncated glycoform displays a greater affinity than the S164A variant that lacks the N162 glycan.

## FcγRIIIa hotspots at the antibody-binding interface

Though high-resolution structural models for the antibody-binding interface are available, the impact of individual residues on binding affinities is not always trivial to predict and thus it is often unclear which residue to target in protein engineering efforts to tune affinity. Our FcγRIIIa mutant screen mapped important residues involved in antibody binding as shown in *Figure 1* to complement one prior study of G129 (*Roberts and Barb, 2018*). Here we identify two hotspots, one formed by W90 and W113 on the FcγRIIIa hinge and another centered on Y132. W90 and W113 are conserved among human low-affinity FcγRs and sandwich the IgG1 Fc P329 residue which is likewise conserved in IgG and IgE (*Sondermann et al., 2000*; *Tamm and Schmidt, 1997*). Previous studies suggest that this proline sandwich interaction is the primary binding interaction in FcγRs, which is consistent with results that mutation abolished binding. The importance of W90 and W113 is also supported by our functional data showing YTS cells expressing the FcγRIIIa W90A or W113A mutants do not exhibit measurable ADCC. Within this same region, I88A bound with twofold lower than V158. Although I88 was not detected in our amino acid selective NMR experiments, nearby V86 and L91 show large perturbations indicating conformational sampling of the interdomain interface, potentially reflecting domain flexibility. In the other hotspot, Y132 is one of many FcγRIIIa residues along the D and E beta strands that interacts with Fc (*Sondermann et al., 2000*). We further probed this region testing Y132, H119, H134, K120, K131, and T122. The Y132S mutation eliminated binding, likely due to removing numerous stabilizing interactions. The IgG1 Fc D265 residue is critical and thought to stabilize the N297 glycan, in which mutation abolishes FcγRIIIa binding (*Baudino et al., 2008*). FcγRIIIa K120 is within distance to interact with D265, possibly playing an important role in Fc binding. We found the K120A mutation reduced binding by fivefold. Other mutations in this region revealed a comparatively minor impact compared to Y132, suggesting Y132 is the critical residue for binding one Fc domain. In addition, the FG loop plays a critical role in Fc binding. While mutation of residues L157 and S160 did not change affinity, K161A lowered affinity by tenfold. The impact of the K161 mutation is interesting

because K161 is not predicted to make direct contact with Fc, although it may contribute to loop stabilization or may transiently contact the IgG1 Fc (*Sondermann et al., 2000*).

## Conclusion

These results indicate FcγRIIIa engineering through substitutions aimed at stabilizing the Fc-bound conformation in the absence of ligand is expected to promote affinity and ADCC. Affinity gains are possible beyond that provided by the tighter-binding V158 allotype, and we demonstrate that relatively small affinity improvements have a substantial impact on ADCC. Finally, we revealed new structural insights into FcγRIIIa that may lead to NK cells engineered to bind antibody through FcγRIIIa with high affinity as a novel strategy to improve immunotherapies.

# Materials and methods

## Materials

All materials were purchased from MilliporeSigma unless otherwise noted.

## SPR

Affinity measurements were performed using the amine coupling strategy with a Biacore T200 (GE Life Sciences) as previously noted (*Kremer and Barb, 2022*).

## Protein expression

Plasmid construction, protein expression, and purification were performed as previously described (*Kremer and Barb, 2022*). Briefly, all FcγRIIIa variants were cloned in the pGen2 vector with an 8x-histidine tag, green fluorescent protein (GFP), and tobacco etch virus (TEV) protease site. These constructs were transiently transfected into HEK293F (Life Technologies) or HEK293S (Gnt1-) cells (*Reeves et al., 2002*) and harvested after 5 days. The collected media was spun down and passed over a Ni-NTA column (QIAGEN) before being stored in 5 mM 3-(N-morpholino) propanesulfonic acid (MOPS), 0.1 M sodium chloride, pH 7.2 buffer.

## Glycosidase digestion

This was purchased from NEB Inc 500 ug of FcγRIIIa-GFP was digested in 5 mM $CaCl_2$, 50 mM sodium acetate pH 5.5 buffer at a 100 uL volume. Digestions of α2-3,6,8 neuraminidase, β1–4 galactosidase S, and β-N-acetylglucosaminidase S (NEB), or all three enzymes were performed overnight at 37°C using one unit each of enzyme. Digestion was analyzed using SDS-PAGE and mass spectrometry.

EndoF1 was expressed with *E. coli* and exchanged into 20 mM sodium phosphate monobasic, 0.1 M KCl, and 0.05 mM 4,4-dimethyl-4-silapentane-1-sulfonic acid in deuterium oxide. 2 uL of 7.5 mg/mL EndoF1 was added to 500 ug of FcγRIIIa in 40 uL and digested in the NMR tube for 2.5 hr at 4°C.

## Selective FcγRIIIa isotope labeling for NMR

FcγRIIIa-GFP was expressed in one of two labeling strategies, either 100 mg/L [15N]-lysine, [15N]-glycine, and [15N]-serine or 100 mg/L [15N]-valine, [15N]-isoleucine, and [15N]-leucine. These labels were supplemented in amino acid-free carbohydrate-free FreeStyle293 medium (Life Technologies) along with 1 g/L glutamine, 100 mg/L of the other amino acids, and 3 g/L glucose. pH then osmolarity were adjusted to 7.2 and 260–280 mOsm/kg, respectively. The final media solutions were confirmed to be at pH 7.2 again before filtration through a sterile 0.2 μm aPES membrane (Fischer Scientific) and storage at 4°C. Following expression in these media, the labeled FcγRIIIa-GFP was TEV protease digested and purified as described (*Kremer and Barb, 2022*).

## NMR spectroscopy of FcγRIIIa N38Q/N74Q/N169Q

The expression plasmid encoding FcγRIIIa N38Q/N74Q/N169Q was generated from the plasmid encoding GFP-FcγRIIIa using site-directed mutagenesis (*Subedi et al., 2014*). HEK293F cells were transfected and protein prepared for NMR as previously described (*Subedi et al., 2014*; *Subedi et al., 2017*). Expression media for individual isotope [15N] labeling was prepared as previously described (*Subedi et al., 2024*). [13C, 15N]-FcγRIIIa N38Q/N74Q/N169Q was expressed in custom Freestyle amino acid and carbohydrate dropout medium (Invitrogen) containing 3 g/L [13C]-glucose

and 100 mg/L of each [$^{15}$N]-labeled amino acid, including alanine, cysteine, aspartate, glutamate, phenylalanine, glycine, histidine, isoleucine, lysine, leucine, methionine, asparagine, arginine, serine, threonine, valine, tryptophan, and tyrosine. Unlabeled proline and glutamine were added a 100 mg/L and 2 g/L, respectively. FcγRIIIa was exchanged into NMR buffer containing 20 mM MOPS, 100 mM potassium chloride, 0.5 mM trimethylsilylpropanesulfonic acid , and 5% deuterium oxide at a final concentration of 100–300 μM.

NMR spectra were collected at a 30°C sample temperature using 700 and 800 MHz AVANCEIII spectrometers (Bruker) equipped with triple resonance cryoprobes. TROSY-based HSQC, HNCA, HNCOCA, and HNCO experiments were selected from the Bruker pulse sequence library. NMR data were processed in NMRPipe and analyzed using NMRViewJ (*Delaglio et al., 1995*; *Johnson, 2004*).

## Lentivirus generation

Plasmids pMD2.G and psPAX2 were gifts from Didier Trono (Addgene plasmid #12259; #12260). Full-length FcγRIIIa variant sequences were cloned under the CMV promoter in pCDH-CMV-MCS-EF1α-copGFP (System Biosciences). These three plasmids were co-transfected into HEK293T cells using Fugene 6 (Promega). Media was exchanged after 3 days and ultimately collected after 5 days total, then passed through a 0.45 μm syringe filter before being stored at –80°C.

## Transduction

$1.5 \times 10^6$ YTS NK cells were cultured in 3 mL of media containing 1.8 mL of previously obtained viral media and 8 μg/mL polybrene (Sigma). The culture plates were spun for 2 hr at $400 \times g$ and then placed in an incubator overnight at 37°C, 5% CO$_2$, and 80% humidity. The following day each well was transferred to 20 mL of fresh media and allowed to expand for another day. GFP signal was used to sort the cells on an S3 bulk sorter (Bio-Rad), with transduction efficiencies as high as 30%.

## Cell culture

YTS cells (a gift from Dr. Mace, Columbia) and Raji cells (ATCC) were grown in RPMI 1640 medium supplemented with 10% FBS, L-glutamine (2 mM), HEPES (10 mM), sodium pyruvate (1 mM), non-essential amino acids (1 mM), and penicillin/streptomycin (50 U/mL) in suspension at 37°C, 5% CO$_2$. The ADCC assay was performed by flow cytometry as described previously (*Rodriguez Benavente et al., 2023*).

## Western blotting

$1.28 \times 10^7$ YTS NK cells were collected and lysed in 100 μL of 1% SDS. Lysates were placed on ice for 1 min and then sonicated for 10 s, repeated three times. Western blotting was performed as previously described (*Shenoy et al., 2021*). Blots were stained with either FcγRIIIa (AF1597; R&D Systems) or GAPDH (NC1955142; R&D Systems). Both primary stainings were followed by anti-goat secondary staining (A32860; Thermo Fisher) before imaging on a LI-COR Odyssey CLx.

Molecular dynamics simulations were performed as previously described using Amber and the Glycam forcefield (*Falconer et al., 2018*). Data were analyzed using VMD.

## Statistical analyses

All statistical analyses were performed with Excel (Microsoft) and Prism 6.09 (GraphPad Software).

## Materials availability statement

YTS NK cell lines developed may be obtained by contacting the corresponding author and through an MTA with the University of Georgia Research Foundation.

## Acknowledgements

We thank Dr. Ganesh Subedi (Iowa State University) for preparing the [$^{13}$C,$^{15}$N]-labeled receptor and collecting the triple-resonance-based NMR experiments and Dr. Ryan Weiss (UGA) for guidance with the lentiviral transduction procedure. AWB was funded by the National Institutes of Health under Award No. U01 AI148114 (NIAID) and from the Biochemistry and Molecular Biology Department at

the University of Georgia, Athens. The content is solely the responsibility of the authors and does not necessarily represent the official views of the National Institutes of Health.

## Additional information

### Funding

| Funder | Grant reference number | Author |
| --- | --- | --- |
| National Institutes of Health | U01AI148114 | Adam W Barb |

The funders had no role in study design, data collection and interpretation, or the decision to submit the work for publication.

### Author contributions

Paul G Kremer, Conceptualization, Investigation, Methodology, Writing – original draft, Writing – review and editing; Elizabeth A Lampros, Conceptualization, Investigation; Allison M Blocker, Investigation, Methodology; Adam W Barb, Conceptualization, Formal analysis, Supervision, Funding acquisition, Writing – original draft, Project administration, Writing – review and editing

### Author ORCIDs

Paul G Kremer  http://orcid.org/0000-0003-3588-0100
Adam W Barb  https://orcid.org/0000-0003-3290-8649

Reviewer #1 (Public review): https://doi.org/10.7554/eLife.100083.3.sa1
Reviewer #2 (Public review): https://doi.org/10.7554/eLife.100083.3.sa2
Author response https://doi.org/10.7554/eLife.100083.3.sa3

## Additional files

### Supplementary files

• Supplementary file 1. Binding affinity measurements for each FcγRIIIa variant.

• Supplementary file 2. Raw ADCC data from the lentivirus transduced YTS cells. V158 = wt; aRTX = afucosylated rituximab; K = kifunensine treatment.

• MDAR checklist

### Data availability

NMR chemical shift assignments are submitted into the BMRB as entry 52304. *Supplementary files 1 and 2* includes individual observations used to generate figures in the manuscript.

The following dataset was generated:

| Author(s) | Year | Dataset title | Dataset URL | Database and Identifier |
| --- | --- | --- | --- | --- |
| Barb A | 2024 | Backbone chemical shifts for the human Fc gamma receptor 3a / CD16a soluble extracellular domain that is glycosylated with two N-glycans at N162 and N45 | https://bmrb.io/data_library/summary/index.php?bmrbId=52304 | Biological Magnetic Resonance Data Bank, 52304 |

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
