## [Editor Report · eLife Assessment]

This study explores the mechanistic link between glycosylation at the N162 site of the Fc gamma receptor FcγRIIIa and the modulation of NK cell-mediated antibody-dependent cytotoxicity. Using innovative isotope labeling strategies and advanced NMR spectroscopy techniques, the authors provide **compelling** evidence of how glycan composition influences receptor stability and immune function. These findings offer **fundamental** insights that may contribute to the development of more effective therapeutic antibodies. The article will be of significant interest to immunologists and researchers focused on therapeutic antibody design.

---

## [Referee Report · Reviewer #1 (Public review)]

Summary:

In this work, the authors continue their investigations on the key role of glycosylation to modulate the function of a therapeutic antibody. As follow up of their previous demonstration on how ADCC was heavily affected by the glycans at the Fc gamma receptor (FcγR)IIIa, they now dissect the contributions of the different glycans that decorate the diverse glycosylation sites. Using a well designed mutation strategy, accompanied by exhaustive biophysical measurements, with extensive use of NMR, using both standard and newly developed methodologies, they demonstrate that there is one specific locus, N162, which is heavily involved in the stabilization of (FcγR)IIIa and that the concomitant NK function is regulated by the glycan at this site.

Strengths:

The methodological aspects are carried out at the maximum level.

Weaknesses:

The exact (or the best possible assessment) of the glycan composition at the N162 site.

---

## [Referee Report · Reviewer #2 (Public review)]

Summary:

The authors set out to demonstrate a mechanistic link between Fcgamma receptor (IIIA) glycosylation and IgG binding affinity and signaling - resulting in antibody-dependent cellular cytotoxicity - ADCC. The work builds off prior findings from this group about the general impact of glycosylation on FcR (Fc receptor)-IgG binding.

Strengths:

The structural data (NMR) is highly compelling and very significant to the field. A demonstration of how IgG interacts with FcgRIIIA in a manner sensitive to glycosylation of both the IgG and the FcR fills a critical knowledge gap. The approach to demonstrate the selective impact of glycosylation at N162 is also excellent and convincing. The manuscript/study is, overall, very strong.

Weaknesses:

After revision, which I feel addressed the minor concerns well, the last comment about significance in the long-term is all that remains. Essentially, it will be important in downstream research to determine whether changes in N162 glycan composition ever occur naturally as a result of some factor(s) that include various disease states, inflammation, age, and so on. The answer (either way) does not diminish the importance of understanding molecular details governing antibody-receptor interactions, but it would be very interesting to know if those glycans are regulated in a way that modulates ADCC activity.

---

## [Author Response]

The following is the authors’ response to the original reviews.

**Public Reviews:**

**Reviewer #1 (Public Review):**
Summary:In this work, the authors continue their investigations on the key role of glycosylation to modulate the function of a therapeutic antibody. As a follow-up to their previous demonstration on how ADCC was heavily affected by the glycans at the Fc gamma receptor (FcγR)IIIa, they now dissect the contributions of the different glycans that decorate the diverse glycosylation sites. Using a well-designed mutation strategy, accompanied by exhaustive biophysical measurements, with extensive use of NMR, using both standard and newly developed methodologies, they demonstrate that there is one specific locus, N162, which is heavily involved in the stabilization of (FcγR)IIIa and that the concomitant NK function is regulated by the glycan at this site.Strengths:The methodological aspects are carried out at the maximum level.Weaknesses:The exact (or the best possible assessment) of the glycan composition at the N162 site is not defined.

We revised the Introduction to include previous findings from our laboratory regarding processing on YTS cells:

“YTS cells, a key cytotoxic human NK cell line used for these studies, express FcγRIIIa with extensive glycan processing, including the N162 site with predominantly hybrid and complex-type glycoforms {Patel 2021}.”

Reviewer #2 (Public Review):Summary:The authors set out to demonstrate a mechanistic link between Fcgamma receptor (IIIA) glycosylation and IgG binding affinity and signaling - resulting in antibody-dependent cellular cytotoxicity - ADCC. The work builds off prior findings from this group about the general impact of glycosylation on FcR (Fc receptor)-IgG binding.Strengths:The structural data (NMR) is highly compelling and very significant to the field. A demonstration of how IgG interacts with FcgRIIIA in a manner sensitive to glycosylation of both the IgG and the FcR fills a critical knowledge gap. The approach to demonstrate the selective impact of glycosylation at N162 is also excellent and convincing. The manuscript/study is, overall, very strong.Weaknesses:There are a number of minor weaknesses that should be addressed.(1) Since S164A is the only mutant in Figure 1 that seems to improve affinity, even if minimally, it would be a nice reference to highlight that residue in the structural model in panel B.

We revised Figure 1B to include the S164 site.

(2) It is confusing why some of the mutants in the study are not represented in Figure 1 panel A. Those affinities and mutants should be incorporated into panel A so the reader can easily see where they all fall on the scale.

We thank the reviewer for this comment. We restructured the Results section to highlight that a primary outcome of the experiment referenced was to map the contribution of interface residues to antibody binding affinity. These data were not previously available, highlighting hotspots at the interface. Figure 1A and B report these results.

We then used a subset of mutations from this experiment, as well as a subset of mutations from an additional library containing mutations proximal to the interface, to build a small library for evaluation using ADCC. The complete binding data for all variants, binding to two different IgG1 Fc glycoforms, is presented in Supplemental Table 1.

T167Y in particular needs to be shown, as it is one of few mutants that fall between what seems to be ADCC+ and ADCC- lines. Also, that mutant seems to have a stronger affinity compared to wt (judged by panel D), yet less ADCC than wt. This would imply that the relationship between affinity and activity is not as clean as stated, though it is clearly important. Comments about this would strengthen the overall manuscript.

We thank the reviewer for this particular insight. We agree that the lack of a clean correlation between ADCC potency and affinity implies additional factors that could have affected these experimental results. We added the following sentence to the discussion.

“Notably, the ADCC potency for those high-affinity variants does not fall cleanly on a line, indicating that other factors affect our observations, which may include organization at the cell surface, changes to glycan composition, or receptor trafficking.”

(3) This statement feels out of place: "In summary, this result demonstrates that the sensitivity to antibody fucosylation may be eliminated through FcγRIIIa engineering while preserving antibody-binding affinity." In Figure 2, the authors do indeed show that mutations in FcgRIIIa can alter the impact of IgG core fucosylation, but implying that receptor engineering is somehow translatable or as impactful therapeutically as engineering the antibody itself deflates the real basic science/biochemical impact of understanding these interactions in molecular detail. Not everything has to be immediately translatable to be important.

We agree and removed the highlighted sentence.

(4) The findings reported in Figure 2, panel C are exciting. Controls for the quality of digestion at each step should be shown (perhaps in supplementary data).

We agree. We added an example of the digestions as Figure S2.

(5) Figure 3 is confusing (mislabeled?) and does not show what is described in the Results. First, there is a F158V variant in the graph but a V158F variant in the text.Please correct this.

Thank you for identifying this typo. We corrected Figure 3.

Second, this variant (V158F/F158V) does not show the 2-fold increase in ADCC with kifunesine as stated.

Thank you for drawing our attention to this rounding error. We revised the text to report a statistically significant 1.4-fold increase.

Finally, there are no statistical evaluations between the groups (+/- kif; +/- fucose).

We provide the p values for +/-fuc and +/- Kifunensine for each YTS cell line in the figure. We did not provide a global comparison of p values that included all cell lines due to some cell lines experiencing a significant change and others not. However, we added the raw data as Supplemental Table 2 should readers wish to perform these analyses.

The differences stated are not clearly statistically significant given the wide spread of the data. This is true even for the wt variant.

We agree that there are points that overlap in this figure between the different treatments. However, our use of the students T-test (two tailed) using three experiments collected on three different days (each with three technical replicates) provides enough resolution to determine the significance of difference of the means for the different treatments. This is, by our estimation, a highly rigorous manner to collect and analyze the data.

(6) The kifunensine impact is somewhat confusing. They report a major change in ADCC, yet similar large changes with trimming only occur once most of the glycan is nearly gone (Figure 2). Kifunensine will tend to generate high mannose and possibly a few hybrid glycans. It is difficult to understand what glycoforms are truly important outside of stating that multi-branched complex-type N-glycans decrease affinity.

Note that Figure 2 does not evaluate the kifunensine-treated glycan, which is mostly Man8 and Man9 structures. In our previous work, these structures likewise provide increased binding affinity (see pubmed ID 30016589). We believe the most important message is that composition of the N162 glycan (removed with the S164A mutation) regulates NK cell ADCC. On cells, we are not able to modulate N162 glycan composition without affecting potentially every other N-glycan on the surface, so we do not have an ADCC experiments that is directly comparable to Figure 2. Thus, this increased ADCC resulting from kifunensine treatment is consistent with previously observed increases in binding affinity measurement.

(7) This is outside of the immediate scope, but I feel that the impact would be increased if differences in NK cell (and thus FcgRIIIA) glycosylation are known to occur during disease, inflammation, age, or some other factor - and then to demonstrate those specific changes impact ADCC activity via this mechanism.

We agree completely. As mentioned in the Introduction, we know that N162 glycan composition varies substantially from donor to donor based on previous work from our lab. Curiously, little variability appeared between donors at the other four Nglycosylation sites. Thus, there is the potential that different NK cell N162 glycan compositions are coincident with different indications. This is an area we are quite interested in pursuing.